# miR-31-5p-Modified RAW 264.7 Macrophages Affect Profibrotic Phenotype of Lymphatic Endothelial Cells In Vitro

**DOI:** 10.3390/ijms232113193

**Published:** 2022-10-29

**Authors:** Aneta Moskalik, Anna Ratajska, Barbara Majchrzak, Ewa Jankowska-Steifer, Krzysztof Bartkowiak, Mateusz Bartkowiak, Justyna Niderla-Bielińska

**Affiliations:** 1Postgraduate School of Molecular Medicine, Collegium Anatomicum, Medical University of Warsaw, 02-004 Warsaw, Poland; 2Department of Pathology, Collegium Anatomicum, Medical University of Warsaw, 02-004 Warsaw, Poland; 3Department of Histology and Embryology, Collegium Anatomicum, Medical University of Warsaw, 02-004 Warsaw, Poland; 4Department of History of Medicine, Medical University of Warsaw, 00-581 Warsaw, Poland

**Keywords:** macrophages, miR-31-5p, lymphatic endothelial cells, fibrosis

## Abstract

Cardiac lymphatic vessel (LyV) remodeling as a contributor to heart failure has not been extensively evaluated in metabolic syndrome (MetS). Our studies have shown structural changes in cardiac LyV in MetS that contribute to the development of edema and lead to myocardial fibrosis. Tissue macrophages may affect LyV via secretion of various substances, including noncoding RNAs. The aim of the study was to evaluate the influence of macrophages modified by miR-31-5p, a molecule that regulates fibrosis and lymphangiogenesis, on lymphatic endothelial cells (LECs) in vitro. The experiments were carried out on the RAW 264.7 macrophage cell line and primary dermal lymphatic endothelial cells. RAW 264.7 macrophages were transfected with miR-31-5p and supernatant from this culture was used for LEC stimulation. mRNA expression levels for genes associated with lymphangiogenesis and fibrosis were measured with qRT-PCR. Selected results were confirmed with ELISA or Western blotting. miR-31-5p-modified RAW 264.7 macrophages secreted increased amounts of VEGF-C and TGF-β and a decreased amount of IGF-1. The supernatant from miR-31-5p-modified RAW 264.7 downregulated the mRNA expression for genes regulating endothelial-to-mesenchymal transition (EndoMT) and fibrosis in LECs. Our results suggest that macrophages under the influence of miR-31-5p show the potential to inhibit LEC-dependent fibrosis. However, more studies are needed to confirm this effect in vivo.

## 1. Introduction

The role of lymphatic vessels (LyVs) is to maintain tissue fluid homeostasis by returning extravasated fluid with electrolytes, waste, and proteins to the blood circulation. Moreover, lymphatic vessels are involved in immune cell trafficking and lipid absorption and transport [1]. There are sparse reports on the ability of lymphatic endothelial cells (LECs) to produce extracellular matrix (ECM), including collagen. Endothelial cells of LyVs are known to produce ECM molecules [2]. It has been demonstrated that LECs express mRNAs for genes coding various molecules, including those that are involved in collagen deposition and processing. These mRNAs are elevated in some pathological settings such as hypoxia, high glucose concentration, and others [2,3]. LyV endothelium is also able to transdifferentiate into the mesenchymal phenotype in vitro resulting in the formation of collagen-producing fibroblasts via endothelial–mesenchymal transition (EndoMT). This process is stimulated by TGF-β [4,5,6]. Dysfunction of lymphatic vessels and insufficient lymphangiogenesis, which are observed, e.g., in cardiovascular disease, lead to the development of edema and impaired inflammatory cell uptake, subsequently resulting in ECM remodeling and interstitial fibrosis augmentation [7]. In this latter scenario, fibrosis has been reported to be accelerated by activated fibroblasts/myofibroblasts that deposit collagen. The cells that are believed to mediate these processes may be macrophages, considered to be highly plastic cells. Macrophages influence fibroblasts, LECs, and other cell types through the release of various mediators, such as cytokines and growth factors [8,9]. Macrophages have been suggested to regulate functions of the surrounding cells by the secretion of microRNAs (miRNAs) in microvesicles [10,11].

miRNAs, endogenous RNAs of about 23 nucleotides in length, perform an important function in negative gene regulation by targeting specific mRNAs for translation repression or degradation [12]. The expression of miRNAs determines many physiological processes, including LEC development and LEC and macrophage functions [13,14,15]. Moreover, changes in miRNA regulation modulate the phenotypes and activities of these cells [16].

Studies involving the impact of miRNAs on tissue homeostasis are limited due to the complexity of interactions between cells and their surrounding environment. Moreover, miRNAs can act differently, depending on the tissue microenvironment and cell type. In our previous paper we demonstrate that miR-31-5p is upregulated in cardiac macrophages isolated from db/db mouse hearts [17]. Db/db mice are used as an animal model of metabolic syndrome (MetS) and exhibit symptoms of heart failure [18]. The myocardium of db/db mice shows a reduced number of lymphatic and blood microvessels, increased fibrosis, and mild inflammation [17]. The molecular mechanisms underlying these changes are not well understood.

Research describing the influence of miRNAs on cardiac tissue homeostasis and pathophysiology is limited; thus, in this study we attempted to evaluate the involvement of miR-31-5p in macrophage–LEC interactions in vitro. We assessed the effect of miR-31-5p on the changes in the gene expression profile of murine macrophage cell line RAW 264.7 and the secretion of factors essential in lymphangiogenesis and fibrosis. The influence of miR-31-5p on the LEC phenotype was investigated directly, by miRNA transfection, as well as indirectly, by culturing C57BL/6 mouse primary dermal LECs with supernatants from miR-31-5p-modified macrophages. Our results show that miR-31-5p modifies the macrophage phenotype in vitro and may exert a protective effect on LECs to maintain their endothelial phenotype and prevent fibrosis.

## 2. Results

### 2.1. mRNA Expression Profile Differs between Db/Db and Control Mouse Myocardium

Experiments comparing mRNA expression profile in the myocardium of db/db and control mice showed that mRNAs for Igf1, Vegfc, Col1a1, Fn1, and Emilin1 were downregulated, and there were no statistically significant differences in levels of mRNA expression for Tgfb1, Smad2, Smad4, Snail1, Snail2, Zeb1, Zeb2, and Cdh2 (Figure 1A). There were no statistically significant differences in the amount of N-cadherin protein (product of Cdh2 gene) between the myocardium of control and db/db mice, assessed by Western blotting. However, there was an upward trend for N-cadherin in the db/db group relative to control (Figure 1B,C).

### 2.2. miR-31-5p Modulates the Phenotype of RAW 264.7 Macrophages

Expression of miR-31-5p in RAW 264.7 macrophages was barely detectable after 24 and 48 h of culture in Dulbecco’s modified medium supplemented with 1% FCS (data not shown); thus, only miR-31-5p mimic was used for macrophage transfection, and miR-31-5p inhibitor was unnecessary. The cells were successfully transfected with miR-31-5p mimic, which was also confirmed with miR mimic positive control that silences the expression of Twf1 gene (data not shown). After 48 h of culture, no morphological changes in RAW 264.7 macrophages were observed under a light microscope (data not shown), but qRT-PCR analysis showed statistically significant downregulation of mRNA expression for Igf1 and upregulation of mRNA expression for Vegfc and Tgfb1 compared with that in both untransfected control and mock transfected cells (Figure 2A). The results were confirmed with enzyme-linked immunosorbent assay (ELISA) by the measurement of selected protein concentrations in the supernatants from miR-31-5p-modified and control RAW 264.7 macrophages (Figure 2B).

### 2.3. miR-31-5p-Modified RAW 264.7 Macrophages Downregulate Factors Related to Endomt, Fibrosis, and ECM Deposition

Supernatants from miR-31-5p-modified RAW 264.7 macrophages were used to stimulate LEC cultures, and supernatants from untransfected macrophages were used in controls. Incubation with supernatants at a 1:4 ratio with Endothelial Cell Medium for 48 h did not affect cell morphology (data not shown) but changed mRNA expression profile in LECs. mRNA expression levels for Cdh2, Smad2, Smad4, Snail1, Snail2, Zeb1, Col1a1, Fn1, and Emilin1 were decreased compared to LECs incubated with supernatant from unmodified RAW 264.7. However, there was no difference for Zeb2 (Figure 3A). There was a downward trend in the amount of N-cadherin protein between the control and stimulated LEC protein extracts as assessed by Western blotting, but there was no statistical significance when optical density was measured (Figure 3B,C).

Since miRNA can be transferred directly from cell to cell, or some miRNA molecules could be present in the supernatants, we also evaluated the direct impact of miR-31-5p on LECs. No significant morphological changes were observed under a light microscope 48 h after transfection (data not shown), and modification of LECs with miR-31-5p did not change the expression of mRNA for Snail1, Snail2, Zeb1, Zeb2, Col1a1, Fn1, and Emilin1 but downregulated mRNA expression for Cdh2 and upregulated mRNA for Smad2 and Smad4 (Figure 4).

## 3. Discussion

In our previous studies we focused on myocardial remodeling, which can be caused by MetS and leads to diabetic cardiomyopathy and heart failure. Pathological changes within the heart include remodeling of blood and lymphatic vessels, perivascular and interstitial fibrosis, as well as mild inflammation [7,17]. The molecular mechanisms underlying these morphological changes are largely unknown and orchestrated by a great number of resident and infiltrating cells, including macrophages [19]. Lymphatic endothelial cells might be a source of fibroblasts via the EndoMT pathway leading to mesenchymal-fibroblastic differentiation under the influence of TGF-β [6] or constitute a direct source of collagen and other ECM molecules under normoxic and hypoxic conditions, as has been recently reported [2].

In our previously published paper, we show that macrophage populations isolated from the myocardium of diabetic db/db mouse exhibit different profile of miRNA expression compared to control [17]. At least some of the affected miRNAs can be involved in angiogenesis, lymphangiogenesis, and fibrosis, and thus their altered levels may be responsible for the morphological changes of the myocardium. On the basis of our results and a thorough literature search, we selected miR-31-5p as a potential candidate for further in vitro studies.

Since the discovery of miRNAs, interest in these molecules as potential diagnostic and therapeutic tools has grown [20]. There is plentiful evidence suggesting that miRNAs regulate cardiac homeostasis and thus may serve as a diagnostic tool and/or therapeutic target [21]. Several miRNAs targeting mRNA-encoding proteins involved in cardiac fibrosis are currently being assessed for their potential use as therapeutic targets, but these approaches are still at an early stage [22]. Therefore, more information is needed to decipher the complicated interactions between miRNAs and the local environment. The function of miR-31 is recognized as potentially important in the progression of cardiovascular diseases. In human atrial fibrillation, the increased level of miR-31 causes arrhythmia [23], and after myocardial infarction it induces detrimental cardiac remodeling [24]. Tincr-miR-31-5p axis targets PRKCE, which is involved in cardiomyocyte hypertrophy [25]. miR-31-5p/155 are upregulated in endothelial cells by inflammatory cytokines and inhibit the eNOS/NO axis, resulting in disruption of vascular homeostasis [26]. In addition, miR-31 takes part in angiogenesis and lymphangiogenesis [27,28]. The miR-31 level is elevated in serum of patients with diabetic microvascular complications [29]. Interestingly, miR-31 inhibits lymphangiogenesis and venous sprouting in embryonic development by targeting PROX1, a major transcription factor responsible for maintaining the phenotype of LECs [28]. It is also suggested that miR-31 could regulate processes related to fibrosis, e.g., miR-31 negatively regulates the fibrogenic pathway of epithelial-to-mesenchymal transition (EMT) by targeting Islet-1 and is a positive regulator of EndoMT [30,31,32].

There are very few studies describing the influence of miR-31 on macrophages. It is known, for example, that miR-31-5p modulates inflammation and oxidative stress in alveolar macrophages in vitro [33] and has an anti-apoptotic effect on macrophages [34]. miR-31 expression is elevated during fibrosis and is stimulated by TGFβ, considered the main signaling molecule involved in this process as demonstrated in the liver [31]. Our previous results showed that miR-31-5p expression was significantly upregulated in macrophages isolated from db/db mouse cardiac tissue, but not in the whole tissue, strongly suggesting that miR-31-5p may be of macrophage origin. Additionally, since miRNAs may alter the metabolism of the cell that produces them, we hypothesized that miR-31-5p may modify the secretome of macrophages and thus has an indirect impact on the cardiac environment, including lymphatic endothelial cells [16].

The lymphatic vessel number in cardiac muscle affected by MetS is decreased, which may result in decreased absorption of interstitial fluid and edema. It is well described that fluid accumulation may, in turn, lead to pathological remodeling of extracellular matrix and fibrosis [7,35]. Morphological changes of the myocardium may be a consequence of the impaired expression of proteins involved in regulation of fibrosis. Therefore, we investigated the mRNA levels for some genes encoding proteins in the whole cardiac tissue and we observed that mRNA for Vegfc, Col1a1, Emilin1, Fn1, and Igf1 were downregulated, but the main regulators of fibrosis and/or EMT—Smad2, Smad4, Zeb1, Zeb2, Snail1, Snail2, Cdh2, and Tgfb1 [36]—were unaffected. Since we were evaluating animals at an age of 21 weeks, i.e., at an advanced stage of disease development, these results may be due to the dynamic changes in mRNA expression within cardiac tissue during the progression of disease or a limited number of cells expressing the above factors. Perhaps db/db mice at 21 weeks of age employ a compensatory mechanism that is expected to reduce collagen deposition. As there is an upward trend in N-cadherin at the mRNA and protein levels in the myocardium of db/db mice (despite not being statistically significant), it cannot be ruled out that EMT or EndoMT mechanisms are initiated by a Smad-independent signaling pathway or were activated earlier on in the life of these animals. Nevertheless, downregulation of Vegfc and Igf1 mRNA may be responsible for a decreased number of lymphatic vessels in the myocardium [37,38]. Moreover, downregulation of mRNA expression for Emilin1 may contribute to pathological changes in the myocardium, particularly lymphatic vessels [39,40], as Emilin 1 takes part in anchoring LECs to the ECM and, therefore, is responsible for the integrity of lymphatics [41,42].

To evaluate the effect of miR-31-5p-transfected macrophages on the LEC phenotype, we transfected RAW264.7 macrophages and assessed the impact of their secretome on LECs in vitro. In this study we demonstrate that miR-31-5p alters RAW264.7 macrophage phenotype and their secretome, lowering IGF-1 and increasing VEGF-C and TGF-β1, which was confirmed at the mRNA and protein levels. Moreover, the secretome from miR-31-5p-modified macrophages downregulates mRNA and protein for factors associated with fibrotic pathways, such as Smad2, Smad4, Snail1, Snail2, Zeb1, Col1, fibronectin 1, emilin 1, and N-cadherin in LECs.

IGF-1 plays an important role both in homeostasis and cardiovascular disease [43]. IGF-1 facilitates endothelial cell migration, proliferation, survival, and tube formation [44,45]. Although many studies have shown that IGF-1 may reduce fibrosis, e.g., in the heart [46,47], in certain conditions IGF-1 has an adverse, profibrotic effect [48]. In this paper we show that supernatant from miR-31-5p-modified macrophages, containing reduced IGF-1 concentration, contributes to downregulation of mRNA expression for Smad2, Smad4, Snail1, Snail2, and Zeb1 in LECs. These genes are involved in fibrotic pathways, e.g., the regulation of EndoMT [49,50,51,52], and may influence cardiac remodeling [53,54]. IGF-1 signaling may regulate expression of some of the above genes. In diabetic kidney disease, elevated levels of IGF-1 result in Snail1 upregulation and profibrotic effects. Blockade of IGF-1/IGF-1R normalizes Snail1 expression and attenuates fibrogenesis [55]. Moreover, IGF-1 promotes EMT by increasing the level of expression of Snail1, Snail2, Twist1, N-cadherin, and vimentin [56]. In prostate cancer cells, IGF-1 upregulates expression of transcriptional factor ZEB1 and also increases expression of fibronectin and N-cadherin [57]. Corneal epithelial cells release IGF-1, affecting corneal fibroblasts in a similar way, and RNA-interference-mediated depletion of IGF-1 prevents this effect [58]. The connection between IGF-1 and ZEB2 was shown in gastric cancer cells, where IGF-1 induced EMT by activation of the PI3K/Akt-GSK-3β-ZEB2 signaling pathway [59]. However, no difference for Zeb2 was detected in our experimental model. IGF-1 may induce αSMA expression and deposition of collagen which is one of the major components of extracellular matrix [60,61]. Interestingly, the blocking of the IGF-1 receptor inhibits collagen deposition [60]. Our experiments demonstrated downregulation of mRNA expression for collagen type I in LECs modified by supernatant from miR-31-5p-transfected macrophages. Furthermore, the level of mRNA for fibronectin 1 was reduced. Inhibition of fibronectin attenuates adverse left ventricular remodeling and fibrosis, preserving cardiac function in a mouse model of heart failure [62]. However, in human lens epithelial cells, IGF-1 prevents TGF-β-mediated fibronectin accumulation [63]. Evaluated collectively, the results suggest that IGF-1 signaling plays an important role in the activation of factors related to fibrosis, and thus inhibition of IGF-1 secretion has an anti-fibrotic potential.

It is known that defective or insufficient lymphangiogenesis contributes to tissue edema and fibrosis [7]; therefore, looking for methods to improve lymphatic vessel function is necessary. Currently, many studies focus on lymphangiogenic therapy by administration of VEGF-C, which is a major prolymphangiogenic mediator [64] and local ECM modulator [65] often released by macrophages [66,67]. Promising results show that treatment with VEGF-C reduces inflammation, tissue edema, improves cardiac and lymphatic functions, and ameliorates hypertension, e.g., after ischemia and reperfusion injury [68], myocardial infarction [69,70], and angiotensin II infusion-induced chronic cardiac dysfunction [71]. Moreover, overexpression of VEGF-C promotes cardiac lymphangiogenesis, reduces macrophage infiltration, and diminishes myocardial fibrosis, thus preserving left ventricular function in hypertensive rats [72]. In our experiments, miR-31-5p-trasfected RAW 264.7 macrophages exhibit an increased secretion of VEGF-C, which in turn reduces mRNA expression of the genes associated with fibrosis in LECs. This supports our hypothesis that miR-31-5p may act protectively in the cardiac environment.

MiR-31-5p also increased production of TGF-β by macrophages in our experiment. TGF-β is a crucial profibrotic cytokine that induces myofibroblast activation and stimulates deposition of ECM proteins [73,74,75]. TGF-β-containing exosomes from cardiac endothelial cells of diabetic mice (streptozotocin-treated) activate fibroblasts for collagen production and aggravate fibrosis, both interstitial and perivascular [76]. Moreover, TGF-β may prompt fibrosis-related lymphangiogenesis through VEGF-C [77,78] and negatively regulates lymphatic regeneration during wound repair [79]. Interestingly, despite increased concentration of TGF-β in the supernatant from macrophages transfected with miR-31-5p, mRNA expression levels for genes encoding proteins involved in fibrosis in LECs decreased. It may suggest that the profibrotic activity of TGF-β is counteracted by IGF-1 and VEGF-C in these cells.

Our findings indicate that the changes in LEC phenotype were caused by modified macrophage supernatant, not by the direct action of miR-31-5p. Only N-cadherin was downregulated after both culturing LECs with the supernatant from miR-31-5p-modified macrophages and miR-31-5p transfection to LECs. N-cadherin is associated with adherens junctions in cells of mesenchymal phenotype, provides cellular interaction, and maintains the structural integrity of tissue [80]. Increased expression of the precursor form of N-cadherin on the surface of myofibroblasts and its abnormal location in damaged tissue of the heart, liver, and lungs may serve as a marker of fibrosis in these organs [81]. There are no studies suggesting that miR-31-5p acts on N-cadherin directly. Therefore, the downregulation of N-cadherin in LECs treated with miR-31-5p in our experimental model is probably an indirect effect.

Our research has some limitations. The mRNA expression of genes from the entire myocardium does not fully reflect the changes occurring locally, i.e., within the cells of the lymphatic vessels of the heart. In addition, the cell culture conditions do not reflect the metabolic syndrome environment but only shed light on the possible effect of miR-31-5p- modified macrophages on LECs. We did not find a target gene for miR-31-5p in this experimental model, and it is uncertain whether all potential factors from the miR-31-5p-modified macrophage supernatant that could alter the phenotype of LECs were identified. However, we have shown for the first time that miR-31-5p may have an influence on the macrophage secretome and therefore may act indirectly on lymphatic endothelial cells, changing their expression profile. Moreover, miR-31-5p may act protectively on LECs, inhibiting, directly and indirectly, N-cadherin expression, which is the main indicator of EMT and EndoMT. More research is needed to evaluate the possible role of miR-31-5p in the pathogenesis of heart failure and its involvement in the regulation of fibrosis.

## 4. Materials and Methods

### 4.1. Animals

The study was performed on male BKS.Cg-Dock7m+/+Leprdb/J mice (db/db); the C57BL/6J strain was used as control. All animal experiments were approved by the First Local Bioethics Committee of the University of Warsaw, Poland and carried out in accordance with EU Directive 2010/63/EU for animal experiments (application accepted 16 November 2016, no. of decision 140/2016). Nine-week-old mice were purchased from Charles River Laboratories (Sant’Angelo Lodigiano, Italy) and kept under specific pathogen-free conditions, with unlimited access to LabDiet^®^ 5K52 (6% fat) chow (Charles River Laboratories, Sant’Angelo Lodigiano, Italy). The animals were sacrificed at the age of 21 weeks by CO2 asphyxiation, and their hearts were isolated for further experiments.

### 4.2. Cell Culture

The murine macrophage cell line RAW 264.7 (American Type Culture Collection—ATCC, Manassas, VA, USA) was maintained in Dulbecco’s modified medium (Gibco, ThermoFisher Scientific, Waltham, MA, USA) supplemented with 10% fetal bovine serum (FCS, HyClone, South Logan, UT, USA) and 1% antibiotic/antimycotic solution (Gibco, ThermoFisher Scientific, Waltham, MA, USA) at 37 °C, 5% CO2. The cells below the 10th passage were used for transfection.

C57BL/6 Mouse Primary Dermal Lymphatic Endothelial Cells (Cell Biologics Inc., Chicago, IL, USA) were cultured in Complete Mouse Endothelial Cell Medium (Cell Biologics Inc., Chicago, IL, USA) with 5% FCS and 1% antibiotic/antimycotic solution in 25 cm^3^ bottles coated with gelatin solution (Cell Biologics Inc., Chicago, IL, USA). For all experiments Primary Dermal Lymphatic Endothelial Cells at the 4th passage were cultured on type I collagen-coated (STEMCELL Technologies, Vancouver, Kanada) plates (1.5 mg/mL acid-soluble type I collagen, buffered with Complete Mouse Endothelial Cell Medium and neutralized with sterile NaOH) at a density of 50,000 cells per well in 24-well plates.

### 4.3. Transfection and Stimulation with Supernatants

For transfection RAW 264.7 macrophages were seeded at a density of 50,000 cells per well in 24-well plates in 500 µL of Dulbecco’s modified medium with 1% FCS. C57BL/6 Mouse primary dermal LECs at the 4th passage were seeded at a density of 50,000 cells per well in 24-well plates onto previously prepared collagen coating in 500 µL of Complete Mouse Endothelial Cell Medium with 1% FCS.

mmu-miR-31-5p mimic (AGGCAAGAUGCUGGCAUAGCUG), miR-1 Positive Control, and miRNA Mimic Negative Control were purchased from mirVanaTM (Invitrogen, Carlsbad, CA, USA). miR-31-5p mimic and control miRNAs were transfected into RAW 264.7 cells and LECs using LipofectamineTMRNAiMAX Transfection reagent (Invitrogen, Carlsbad, CA, USA) following the manufacturer’s instruction. In brief, for one well, a mixture of 10 pmol of miRNA mimic or control, 1.5 µL of Lipofectamine RNAiMAX, and 50 µL of Optimem (Gibco, Thermo Fisher Scientific, Waltham, MA, USA) was prepared and added into 500 µL of culture medium. After 24 h, the medium with miRNA was carefully removed and replaced with 500 µL of fresh culture medium. After 48 h, the transfected cells and supernatants were collected separately for further analysis. During culture, the cells were also observed under inverted light microscope (Zeiss PrimoVert Inverted Phase Contrast Microscope, Carl Zeiss, Dresden, Germany), and photos were taken.

C57BL/6 Mouse Primary Dermal Lymphatic Endothelial Cells were also cultured with the supernatant from transfected RAW 264.7 macrophages in Complete Mouse Endothelial Cell Medium with 1% FCS at a 1:4 ratio at 37 °C, 5% CO_2_. After 48 h, photos of the culture were taken, and the cells were collected for further analysis.

### 4.4. Total RNA Isolation, Reverse Transcription (RT) and Realtime PCR

Thirty-milligram tissue samples taken from the mouse myocardia were transferred to Lysis Buffer and homogenized. Cultured cells were washed with ice cold PBS and suspended in Lysis Buffer. Total RNA was isolated with NucleoSpin ^®^RNA II kit (Macherey-Nagel, Düren, Nordrhein-Westfalen, Germany) according to the manufacturer’s protocol. The concentration and purity of RNA were estimated with a NanoDrop spectrophotometer, and 500 ng of total RNA was reverse transcribed with the High-Capacity RNA-to-cDNA kit (Applied Biosystems, ThermoFisher Scientific, Waltham, MA, USA), according to the manufacturer’s protocol. cDNA was stored at −20 °C for further experiments. Gene expression was measured via relative quantificationRQ) with a comparative CT method. Real-time PCR was performed with Abi Prism 7500 (Applied Biosystems, ThermoFisher Scientific, Waltham, MA, USA) in 96-well optical plates. Each sample was run in triplicate, mouse Gapdh (Mm99999915_g1) was used as an endogenous control. Twf1 (Mm01598982_g1) was a gene silenced by miR-1 Positive Control. TaqMan Gene Expression Assays (Thermo Fisher Scientific, Waltham, MA, USA) were used to measure mRNA for selected genes in RAW 264.7 macrophages, LECs transfected with miR-31-5p, LECs modified by macrophage supernatants and in the myocardium of db/db and control mice: Cdh2: Mm00483208_m1; Col1a1: Mm00801666_g1; Emilin1: Mm00467244_m1; Fn1: Mm01256744_m1; Igf1: Mm00439560_m1; Smad2: Mm00487530_m1; Smad4: Mm03023996_m1; Snail1: Mm00441533_g1; Snail2: Mm00441531_m1; Tgfb1: Mm01178820_m1; Vegfc: Mm00437310_m1; Zeb1: Mm00495564_m1; Zeb2: Mm00497196_m1.

The reactions were run with TaqMan Universal Master Mix (Applied Biosystems, ThermoFisher Scientific, Waltham, MA, USA), primer sets, an MGB probe, and cDNA template (5 ng per reaction) in universal thermal conditions: 10 min at 95 °C and 40 cycles of 15s at 95 °C and 1 min at 60 °C). The data was analyzed with sequence detection software version 1.4 (Applied Biosystems, ThermoFisher Scientific, Waltham, MA, USA).

### 4.5. Enzyme-Linked Immunosorbent Assay (ELISA)

IGF1, VEGF-C, and TGF-β1 concentrations were measured in supernatants from transfected RAW 264.7 macrophages with Mouse Insulin-Like Growth Factor 1 (IGF1) ELISA Kit (MyBioSource, San Diego, CA, USA, cat. No. MBS450396), Mouse Vascular Endothelial Growth Factor C (VEGFC) ELISA Kit (MyBioSource, San Diego, CA, USA, cat. no MBS2701385), and TGFB1 ELISA Kit (Mouse) (Aviva Systems Biology, San Diego, CA, USA, cat. No. OKBB00255) according to manufacturers’ protocols. Measurements were taken with FLUOstar Omega Microplate Reader (BMG Labtech, Ortenberg, Germany).

### 4.6. Western Blotting

For protein detection the cells or tissue samples were lysed in RIPA buffer (50 mM TRIS-HCl (pH 7.4), 150mM NaCl, 1% NP-40, 0.5% sodium deoxycholate, and 0.1% SDS with 1% Triton X-100, all from Sigma Aldrich, Saint Louis, MO, USA), containing protease inhibitors (Roche, Basel, Switzerland). Additionally, cells cultured on collagen matrices before lysis were washed with ice-cold PBS and incubated with 0.1% collagenase type I (Sigma Aldrich, Saint Louis, MO, USA) solution in FCS-free EMC medium at 37 °C for 1 h, washed with PBS, and suspended in RIPA buffer. Protein concentrations were determined with a BCA™ protein assay kit (Pierce, Thermo Fisher Scientific, Waltham, MA, USA), according to the manufacturer’s protocol; 20 µg of total protein was mixed with 2× Laemmli Sample Buffer (4% SDS; 20% glycerol; 0.004% bromphenol blue; 0.125M Tris-Cl, pH 6.8; 10% 2-mercaptoethanol, all from Sigma Aldrich, Saint Louis, MO, USA), boiled, and separated by 10% SDS-PAGE in running buffer (25 mM Tris, 192 mM glycine, 0.1% SDS, all from Sigma Aldrich, Saint Louis, MO, USA) at 90 V. Afterwards proteins were transferred to polyvinylidene difluoride (PVDF) membrane with the semidry TransBlot cell (Bio-Rad, Hercules, CA, USA) in transfer buffer (39 mM glycine, 48 mM Tris base, 20% Methanol, all from Sigma Aldrich). After transfer and prior to antibody application, the membrane was blocked with a 5% semi-skimmed milk solution (Bio-Rad, Hercules, CA, USA) in TBS buffer in RT for 1 h and washed carefully with TBS and TTBS. For detection of N-cadherin, the membrane was incubated overnight at 4 °C with mouse monoclonal anti-N-cadherin and anti-GAPDH antibodies (Thermo Fisher Scientific, Waltham, MA, USA, AM4300 and Invitrogen, Thermo Fisher Scientific, Waltham, MA, USA, MA1-91128, respectively) diluted in TBST at a ratio of 1:250. For signal detection a BM Chemiluminescence Western Blotting Kit (Roche, Basel, Switzerland) was used, according to the manufacturer’s protocol. Membranes were scanned with Syngene G:BOX Chemi Imaging System (Syngene, Cambridge, UK) immediately after the substrate was added.

### 4.7. Statistical Analysis

RT-PCR and ELISA test data were analyzed with GraphPad Prism 9 (GraphPad Software, San Diego, CA, USA). The normality of the distribution was assessed by the Shapiro–Wilk test. The *t*-test or Mann–Whitney test were used, depending on data distribution. Results were considered statistically significant at a *p*-value ≤ 0.05.

## Figures and Tables

**Figure 1 ijms-23-13193-f001:**
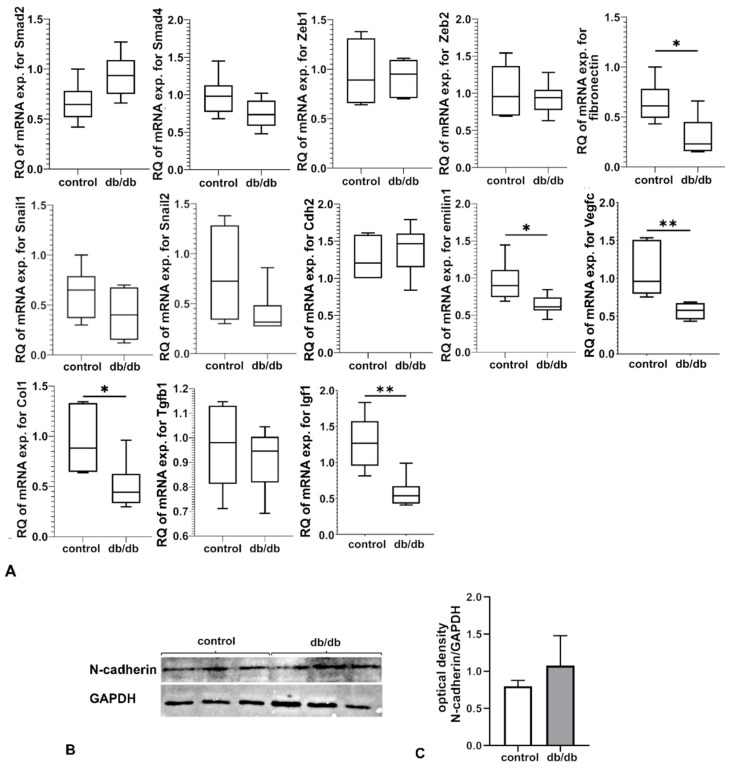
(**A**) Expression of mRNA for selected genes in the myocardium of db/db and control mice. Gene expression was measured via relative quantification (RQ) with the comparative CT method. The results shown are obtained from tissue samples from six animals (n = 6) with technical duplicates. For the statistical analysis of the results, the Student’s *t*-test (* *p* ≤ 0.05; ** *p* < 0.01) was used. (**B**) Assessment of the amount of N-cadherin protein in the myocardium of control and db/db mice with Western blotting. (**C**) The WB results obtained from tissue samples from three animals (n = 3) are shown as the ratio of the optical density of N-cadherin to the optical density of GAPDH. The Student’s *t*-test was used for the statistical analysis of the results.

**Figure 2 ijms-23-13193-f002:**
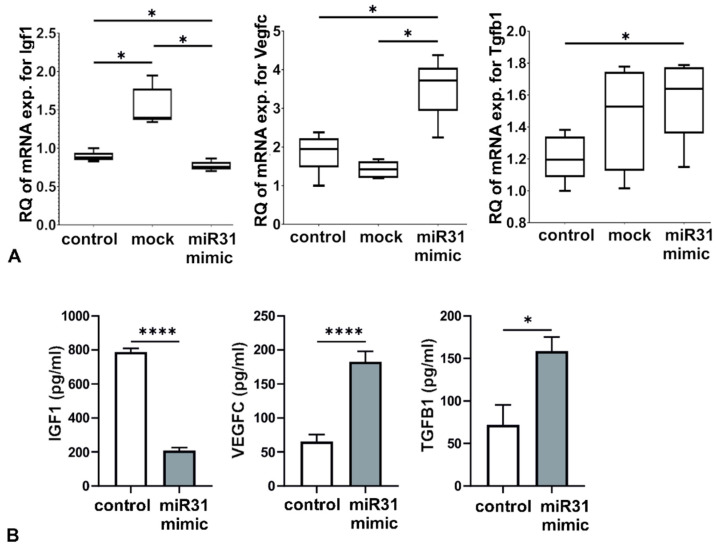
(**A**) Assessment of mRNA expression for selected genes in RAW 264.7 macrophages transfected with miR-31-5p. Gene expression was measured via relative quantification (RQ) with a comparative CT method. The results shown are obtained from five independent experiments (n = 5) with technical duplicates. For the statistical analysis of the results, the Student’s *t*-test was used (* *p* ≤ 0.05). (**B**) Concentrations of selected proteins in the supernatant of RAW 264.7 macrophage culture. The results obtained from five independent experiments (n = 5) with technical duplicates are presented as mean ± SEM. The Student’s *t*-test was used for statistical analysis (* *p* ≤ 0.05; **** *p* < 0.0001).

**Figure 3 ijms-23-13193-f003:**
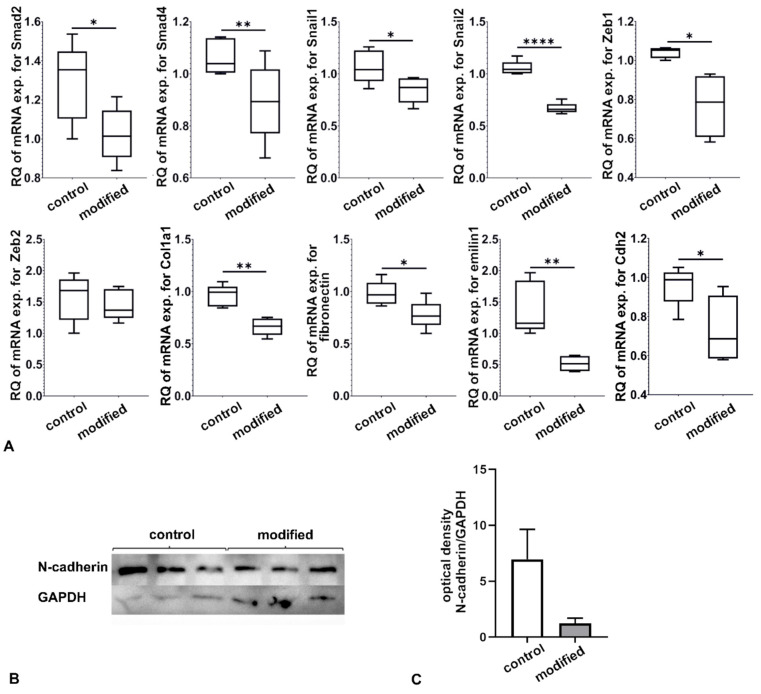
(**A**) Expression of mRNA of selected genes in LECs after incubation with modified RAW 264.7 macrophage supernatant. Gene expression was measured via relative quantification (RQ) with a comparative CT method. The results shown are obtained from five independent experiments (n = 5) with technical duplicates. For the statistical analysis of the results (n = 6), the Student’s *t*-test (* *p* ≤ 0.05; ** *p* < 0.01; **** *p* < 0.0001) was used. (**B**) Assessment of the amount of N-cadherin protein in cultured LECs detected with Western blotting. (**C**) The WB results obtained from three independent experiments (n = 3) are shown as the ratio of the optical density of N-cadherin to the optical density of GAPDH. The Student’s *t*-test was used for the statistical analysis of the results.

**Figure 4 ijms-23-13193-f004:**
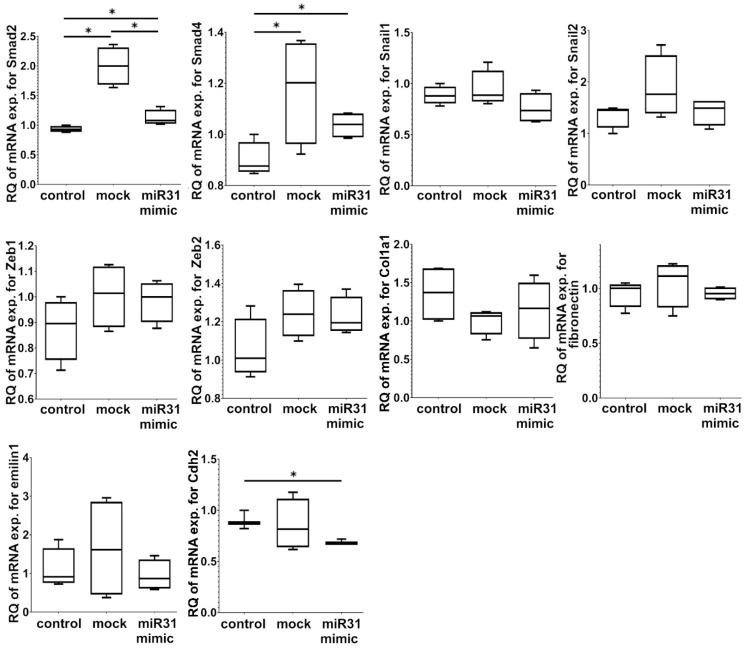
mRNA expression of selected genes in LECs transfected with miR-31-5p. Gene expression was measured via relative quantification (RQ) with a comparative CT method. The results shown are obtained from five independent experiments (n = 5) with technical duplicates. The Student’s *t*-test (* *p* ≤ 0.05) was used for the statistical analysis of the results.

## Data Availability

The datasets generated during and/or analyzed during the current study are available from the corresponding author.

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
