# Peer review of "miR-31-5p-Modified RAW 264.7 Macrophages Affect Profibrotic Phenotype of Lymphatic Endothelial Cells In Vitro"

_ijms, 2022, doi:10.3390/ijms232113193_

Round 1
Reviewer 1 Report
Non-coding RNAs such as microRNAs (miRNAs) have recently received attention as powerful regulators of transcripts and intracellular signaling in norm and in many disease conditions including myocardial fibrosis. Therefore miRNAs are promising as novel drugs or drug targets, as well as potential diagnostic biomarkers. In this context, the submitted manuscript is of interest and timely. It presents new data obtained in mouse cells cultured in vitro showing the direct effect of miRNA-31 transfection on transcriptome and secretome of macrophages. In turn, the supernatant of macrophages transfected with miR-31 was found to affect primary lymphatic endothelial cells (LECs), causing downregulation of genes involved in LEC-dependent fibrosis. The results demonstrate that miR-31 may play a protective role for LECs preventing their endothelial-to-mesenchymal transition and myocardial fibrosis. The article is generally well-written and structured, but several major and minor issues should be addressed prior to publication.
1. In the transfection experiments shown in Fig. 2 and 4, the authors did not present the results of a scrambled sequence as a negative control. Such a control for non-specific effects of miRNA transfection is required. In the method section there is information that only positive control mimic miRNA was used. It needs explanation.
2. The description of data presented in Fig. 1-4 do not specify how many biological and how many technical repeats were taken for statistical analysis.
3. Each strand of the miRNA duplex (guide and passenger) possesses a unique seed sequence defining the strand specificity toward its targets and related signaling pathways. This specific identification (-3p, -5p) is provided for miRNA-31 only in the Method section (miR-31-5p), while it should be used in the whole article for the clarity.
4. A short paragraph on current strategies towards potential application of miRNAs and their mimics or inhibitors as therapeutics would strengthen the Discussion section. This perspective is interesting the context of the presented results, especially that there are good recent reviews on this topic (e.g. 1. New Insights into the Functions of MicroRNAs in Cardiac Fibrosis: From Mechanisms to Therapeutic Strategies. Genes (Basel). 2022, doi: 10.3390/genes13081390; 2. MicroRNAs as therapeutic targets in cardiovascular disease. J Clin Invest. 2022, doi: 10.1172/JCI159179; Antisense oligonucleotides for Alzheimer's disease therapy: from the mRNA to miRNA paradigm. EBioMedicine. 2021, doi: 10.1016/j.ebiom.2021.103691).
5. The article needs some minor editorial corrections, e.g. in the Abstract, “Western Blot” is incorrect, should be changed to “Western blotting” or “immunoblotting” (name of the technique).
Author Response
Thank you very much for valuable comments. Please find below our answers and comments:
- In the transfection experiments shown in Fig. 2 and 4, the authors did not present the results of a scrambled sequence as a negative control. Such a control for non-specific effects of miRNA transfection is required. In the method section there is information that only positive control mimic miRNA was used. It needs explanation.
Response: Thank you for this remark. The results of a scrambled sequence as a negative control have been added to Figures 2 and 4 in the revised manuscript, as you have suggested. Also, information about mock (negative) control was added in the Methods section.
- The description of data presented in Fig. 1-4 do not specify how many biological and how many technical repeats were taken for statistical analysis
Response: Thank you for this crucial comment. We added the relevant information on the number of samples and the number of repeats to Figure legends 1-4, accordingly to your comment.
- Each strand of the miRNA duplex (guide and passenger) possesses a unique seed sequence defining the strand specificity toward its targets and related signaling pathways. This specific identification (-3p, -5p) is provided for miRNA-31 only in the Method section (miR-31-5p), while it should be used in the whole article for the clarity.
Response: We agree that the information about the miRNA strand needs to be specified. Therefore, we specified relevant miRNA strands along the entire revised manuscript wherever it was possible. We must, however, admit that not all authors in the cited references specify the type of strand therefore, it was not possible to make corrections in all instances where miRNA strands were mentioned in the revised manuscript.
- A short paragraph on current strategies towards potential application of miRNAs and their mimics or inhibitors as therapeutics would strengthen the Discussion section. This perspective is interesting the context of the presented results, especially that there are good recent reviews on this topic (e.g. 1. New Insights into the Functions of MicroRNAs in Cardiac Fibrosis: From Mechanisms to Therapeutic Strategies. Genes (Basel). 2022, doi: 10.3390/genes13081390; 2. MicroRNAs as therapeutic targets in cardiovascular disease. J Clin Invest. 2022, doi: 10.1172/JCI159179; Antisense oligonucleotides for Alzheimer's disease therapy: from the mRNA to miRNA paradigm. EBioMedicine. 2021, doi: 10.1016/j.ebiom.2021.103691).
Response: Thank you for this remark and reference suggestions. We added a short paragraph emphasizing the potential use of miRNAs as a therapeutic and diagnostic tool.
- The article needs some minor editorial corrections, e.g. in the Abstract, “Western Blot” is incorrect, should be changed to “Western blotting” or “immunoblotting” (name of the technique).
Response: The article was corrected according to your suggestions and additionally was thoroughly checked by a specialist in English writing.
Reviewer 2 Report
This manuscript is interesting although it is mainly based on the study of only one (murine) macrophages cell line in vitro. In addition, the manuscript is flawed due to experimental conditions that are not always very clear.
detail points:
Figure 1, it is not clear what the Mann-Withney or student'T-Test are adressed. Untransfected controls should be replaced by mock transfected controls (e.g. a scrambled miRNA).
Figure 1B, the quality of the western blot is quite poor. I have some concerns regardind quantitation if the presented WB is representative.
Figure 2 A. I do't see any obvious difference between the control and transfected cells. I think that under theses circumstances, this micrograph is dispensable.
Figure 3A: as above, it seem that this panel is dispensiable;
the quality of WB is rather poor.
Figure 3D, I don't understand why, according to the bar graphs there is noo significant difference. This leads to wonder if this WB was the best to represent results ?
Figure 4, as for or other figures control cells correspond to untransfected cells.
Discussion, to me the disussion is far too long and should be shortened.
Author Response
Thank you very much for valuable comments. Please find below our answers and comments:
This manuscript is interesting although it is mainly based on the study of only one (murine) macrophages cell line in vitro. In addition, the manuscript is flawed due to experimental conditions that are not always very clear.
detail points:
Figure 1, it is not clear what the Mann-Withney or student'T-Test are adressed. Untransfected controls should be replaced by mock transfected controls (e.g. a scrambled miRNA). Figure 4, as for or other figures control cells correspond to untransfected cells.
Response: Thank you for pointing out this important statistical detail. In fact, we used only Student’s T-test, not Mann-Whitney test for the statistical evaluation of the data presented in Figure 1. Therefore, we specified the use of Student’s T-test in the legend of Figure 1. We also corrected and specified the use of both tests in the legends of Figures 2-4.
The results of a scrambled sequence as a negative control have been added to Figures 2 and 4 in the revised manuscript, as you have suggested. We also left untransfected controls on the panels, since they show a threshold of mRNA expression for each molecule in unstransfected cells.
Figure 1B, the quality of the western blot is quite poor. I have some concerns regardind quantitation if the presented WB is representative.
Response: We agree that Western blot results are of poor quality; however, the results are visible and sufficient to measure (optical density). To confirm our point, we enclose our raw data on OD measurement for your information.
|
gapdh |
ncadh |
ncadh/gapdh |
control 1 |
5923.062 |
4035.447 |
0.68131095 |
control 2 |
5048.648 |
4778.305 |
0.946452397 |
control 3 |
5891.305 |
4539.355 |
0.770517738 |
db/db heart 1 |
9579.841 |
3711.891 |
0.387468957 |
db/db heart 2 |
4624.305 |
4954.184 |
1.071335909 |
db/db heart 3 |
2617.941 |
4647.527 |
1.775260405 |
Figure 2 A. I do't see any obvious difference between the control and transfected cells. I think that under theses circumstances, this micrograph is dispensable.
Figure 3A: as above, it seem that this panel is dispensable;
Response: We agree with your comment; therefore these panels were removed.
the quality of WB is rather poor.
Figure 3D, I don't understand why, according to the bar graphs there is noo significant difference. This leads to wonder if this WB was the best to represent results ?
Response: We agree that Western blot results are of poor quality; however, the results are visible and sufficient to measure (optical density). To confirm our point, we enclosed our raw data on OD measurement for your information. The reason was due to difficulties in separating LECs from the collagen gel matrix when collecting cells for Western blot analysis. Collagen gel remnants always gave a little background in our Western blot results. Since LECs grow only on collagen gel, and do not grow on gelatin, Matrigel, or fibrin, we were forced to use collagen matrix and were unable to replace collagen with another cell culture scaffold such as Matrigel or fibrin. The lack of differences between groups was due to a relatively large dispersion of the results, but we decided to show this result because there is a clearly visible tendency for N-cadherin downregulation.
|
gapdh |
ncadh |
ncadh/gapdh |
control 1 |
2675.00 |
31380.70 |
11.73 |
control 2 |
2704.83 |
18074.19 |
6.68 |
control 3 |
3698.43 |
9040.63 |
2.44 |
supernatant 1 |
9433.46 |
12017.68 |
1.27 |
supernatant 2 |
21247.32 |
8368.24 |
0.39 |
supernatant 3 |
8242.72 |
16645.97 |
2.02 |
Discussion, to me the disussion is far too long and should be shortened.
Response: According to your suggestions we edited the Discussion section and shortened the text.
Round 2
Reviewer 2 Report
Revisions of the manuscript are statisfactory.